# Synthesis, characterization, and *in vivo* safety evaluation of propylated *Dioscorea abyssinica* starch

**Yonas Brhane[1], Tsige Gebre-Mariam[1]\*, Anteneh Belete[1,2]**

**1** Department of Pharmaceutics and Social Pharmacy, School of Pharmacy, College of Health Sciences, Addis Ababa University, Addis Ababa, Ethiopia, **2** Center for Innovative Drug Development and Therapeutic Trials for Africa (CDT AFRICA), College of Health Sciences, Addis Ababa University, Addis Ababa, Ethiopia

\* tsige.gmariam@aau.edu.et

**Data Availability Statement:** All relevant data are within the paper and its Supporting Information files.

**Funding:** One of the authors (Y.B.) would like to thank Addis Ababa university and the Center for

## Abstract

The use of starch, a natural polymeric material, and derivatives thereof is based on its adhesive, thickening, gelling, swelling, and film-forming properties, as well as its ready availability. The objective of this research work is to develop an effective propylated *Dioscorea abyssinica* starch (PDAS) as a hydrophobic excipient for pharmaceutical applications with a reasonable price. This paper reports on the synthesis, characterization, and *in vivo* safety evaluation of PDAS. *Native Dioscorea abyssinica* starch (NDAS) was modified to its propylated form with propionic anhydride and characterized. Crystallinity, morphological structure, thermal behavior, solubility, and safety of PDAS were evaluated using x-ray diffraction, SEM, thermogravimetric, gravimetric, and toxicity studies, respectively. Propionyl content and degree of substitution (DS) of starch increased significantly ($p < 0.05$) with an increase in reaction time and temperature. Propionyl content and DS of starch increased significantly ($p < 0.05$) with a decrease in the ratio of starch to pyridine and starch to propionic anhydride in the reaction medium. FTIR spectra of PDAS indicated that hydroxyl groups participated in the propylation reaction. X-ray diffraction results showed that the chemical modification destroyed the crystalline structure of the NDAS. SEM of NDAS showed a rounded shape which became irregular after propylation. Thermogravimetric curves revealed that all the PDAS samples decomposed at higher temperatures than their native counterparts. At higher DS, swelling power and solubility in an aqueous environment significantly ($p < 0.05$) decreased below that of the native starch. PDAS with high DS, were soluble in organic solvents at room temperature. But PDAS with lower DS didn't dissolve in all types of organic solvents used. PDAS (DS = 2.842) in distilled water did not produce adverse effects in rats. Based on the results obtained, it can be concluded that PDAS can be considered as a generally safe excipient and fulfills the physicochemical properties of a hydrophobic excipient.

## Introduction

Starch and its derivatives are the most utilized additives in the production of various pharmaceutical formulations. A few of the Ethiopian plants which have been studied and proven to

Innovative Drug Development and Therapeutic Trials for Africa (World Bank Credit No. 5794-ET, Project No. P151847-IDA57940) for sponsoring the work. The funders of this study had no role in study design, data collection and analysis, decision to publish, or preparation of the manuscript.

**Competing interests:** The authors have declared that no competing interests exist.

have starch for pharmaceutical applications include Enset (*Ensete ventricosum*) [1]; Ethiopian Yam (*Dioscorea abyssinica*) [2]; Ethiopian Potato (*Plectranthus edulis*) [3]; Anchote (*Coccinia abyssinica*) [4]; Cassava (*Manihot esculenta*) [5] and Ethiopian oat (*Triticum decocum*) [6].

*Dioscorea abyssinica (Fam. Dioscoreaceae)* is a climber with an herbaceous stem and a big tuber that twines to the right. It is grown in Ethiopia's south, west, and southwest highlands during the rainy season. *Dioscorea* tubers have long been utilized as food due to their high starch content. *Dioscorea* is frequently referred to as "boyna" among the local community. On a dry weight basis, starch is the most important component of *Dioscorea* tuber, accounting for around 80% of the dry weight. The boyna starch has the following approximate compositions: 0.1 percent ash, 0.5 percent protein, 1 percent fat, and 98.4 percent starch. The X-ray diffraction pattern of Boyna starch is characteristic of B-type, with an apex close to 17˚ (2θ) [2].

Starch is a biocompatible, biodegradable, nontoxic polymer widely occurring in nature as the major polysaccharide storage in higher plants. The limitation of native starch, like unpredictable viscosity after heating, excessive propensity for retrogradation, absence of good flow properties, thermal degradation, and insolubility in aqueous media restrict its use in the manufacturing industries [7,8]. Modification of starch, changing the structure of starch, is done by substitution of the hydrogen bonds in the glucose residues [9]. The presence of numerous hydroxyl groups in native starch can be used to advantage for anchoring hydrophobic side chains to address the aforementioned limitations [10–18]. The hydroxyl groups of the glucose residues in starch exhibit reactivity typical for alcohols and could undergo conversion into ethers by various alkylating agents such as methyl, ethyl, propyl, and other alkyl halides, and into esters by propionic anhydride as shown in **Fig 1.** [19–22].

Acute and subacute toxicity studies are conducted to know the adverse effects of a substance on human or animal health. Significant changes in body weights are a clear indication of damage caused by the test substance [23].

**Fig 1. The chemical reaction of starch and propionic anhydride.**

The production of hydrophobically modified excipient from local and abundantly available sources, like starch, improves overall performance and increases the use of starch in various applications such as coating, controlled release, enhanced cellular delivery of drugs, and diagnostic agents [11,19,22,24]. Also, it adds new information to the scientific community as far as excipients are concerned. This paper reports on the synthesis, characterization, and *in vivo* safety evaluation of PDAS as a hydrophobic excipient.

## Materials and methods

### Materials

At its maturity age (7 months after planting), *Dioscorea abyssinica* fresh tubers were collected from Saula, Gamogofa Zone, Southern Ethiopia. Sodium metabisulphite 99.5% (Sigma Chemicals Ltd., St. Louis, USA), propionic anhydride 98% (Sigma Aldrich, Germany), hydrochloric acid 37% (Guangzhou Jinhuanda Chemical Reagent Co. Ltd, China), NaOH 99.9% (HiMedia, Mumbai, India), and isopropanol 99% (Sigma-Aldrich, Germany) were used as received. Other chemicals, namely, pyridine99.8%, methanol 98%, ethanol 96.5%, KOH 99.9%, and phenolphthalein indicator 1% were obtained from Fine Chemicals Limited, Mumbai, India and were used as received.

### Methods

**Starch extraction from *Dioscorea abyssinica* tuber.** Starch was extracted and purified following the procedure described by Gebre-Mariam and Schmidt [2]. *Dioscorea abyssinica* tubers (1 kg) were washed, trimmed to remove defective parts, and peeled. Immediately after peeling, the tuber flesh was chopped and kept in a large amount of distilled water containing 0.075% (w/v) of sodium metabisulphite. The slurry was then allowed to sediment for 24 h, and the supernatant was decanted. The slurry was repeatedly treated with sodium metabisulphite solution until the supernatant was clear. To remove cell debris the suspension was then passed through fine muslin and the translucent suspension was collected and allowed to settle. The sediment was washed several times with distilled water by filtering and re-suspending until the wash water was clear and free of suspended impurities. The resulting starch was air-dried at room temperature, milled and sieved through a fine sieve (224μm), and stored in an airtight container for further use. The percent yield was then calculated.

**Synthesis of PDAS.** Table 1 provides the experimental conditions for the propylation of starch. A dried starch sample (20 g) was taken in a reaction flask followed by the addition of

Table 1. Reaction conditions for the synthesis of PDAS.

| Reaction condition | Reaction Time (h) | Reaction Temperature (˚C) | Starch: pyridine (g/mL) | Starch: propionic anhydride (g/mL) |
|---|---|---|---|---|
| A | 3 | 75 | 1:5 | 1:8 |
| B | 6 | 75 | 1:5 | 1:8 |
| C | 12 | 75 | 1:5 | 1:8 |
| D | 24 | 75 | 1:5 | 1:8 |
| E | 24 | 50 | 1:5 | 1:8 |
| F | 24 | 60 | 1:5 | 1:8 |
| G | 24 | 75 | 1:1 | 1:8 |
| H | 24 | 75 | 1:2.5 | 1:8 |
| I | 24 | 75 | 1:10 | 1:8 |
| J | 24 | 75 | 1:5 | 1:1 |
| K | 24 | 75 | 1:5 | 1:2 |
| L | 24 | 75 | 1:5 | 1:4 |

pyridine (at 1:1, 1:2.5, 1:5, and 1:10 ratio) as a catalyst. The flask was then heated to 90˚C for 2 h to preactivate the starch. A reflux condenser was used to prevent the loss of organic liquid. After the preactivation for 2 h, the reaction mixture was cooled to (50, 60, and 75˚C). Then the activated starch was mixed with propionic anhydride (at 1:1, 1:2, 1:4, and 1:8 ratio) and the reaction continued for (3, 6, 12, and 24 h) to ensure reaction equilibrium. The content of the reaction mixture was precipitated by adding 100 mL of isopropanol. The product was filtered and washed with methanol three times. Finally, the product was dried in an oven (Kottermann® 2711, Germany) at 70˚C for 24 h [25].

**Characterization of PDAS.** *Determination of propionyl content and DS.* Propionyl content and DS were determined by the titration method. Accurately weighed propylated starch (1.0 g) was transferred to a 250 mL flask and 50 mL of 75% ethanol was added and stirred at 50˚C for 30 min at 40 rpm on an electrically heated hot plate using a magnetic stirrer and cooled to room temperature. This was followed by the addition of 0.5M KOH (40 mL) while swirling the content of the flask. The flask was stoppered and allowed to stand for 72 h with stirring at 40 rpm for 10 min every 2 h for complete saponification. The excess alkali was then back titrated with 0.5M HCl using a phenolphthalein indicator. The solution was allowed to stand for 2 h and additional alkali which leached from the sample was titrated. In parallel, a blank was titrated. The propionyl content and the DS were calculated from Eqs (1) and (2), respectively [25].

$$\%\text{propionyl content} = \frac{[(\text{blank, mL} - \text{sample, mL}) \times \text{molarity of HCl} \times 0.057 \times 100]}{\text{sample weight, g}} \quad (1)$$

$$\text{DS} = \frac{(162 \times \text{propionyl\%})}{[57 \times 100 - ((57-1) \times \text{propionyl\%})]} \quad (2)$$

Where 162 is the molecular weight of anhydro glucose units and 57 is the formula weight of the propionyl group.

*Determination of Fourier Transform Infrared (FTIR) spectra.* FTIR spectra, with transmittance mode, of NDAS and PDAS samples, were obtained at room temperature by FTIR spectrophotometer (PerkinElmer, Spectrum Two DTGS, UK) using the KBr disc technique. FTIR spectra were recorded at a resolution of 4 cm$^{-1}$ and wave numbers ranging between 4000 and 400 cm$^{-1}$ [26]. Data acquired were plotted in Origin Pro 8.5.1 software.

*X-ray diffraction studies.* X-ray powder diffraction of the starch samples was taken with an X-Ray diffractometer (Philips PW 3710; XPERT-PRO PW 3710, Philips, Japan) operating in the 2θ modes. A Cu target tube operated at a power setting of 40 kV (30 mA) in the range of 5˚ to 60˚ of 2θ with a single crystal graphite monochromator equipped with a microprocessor to analyze peak position and intensities were utilized. A standard polycrystalline silicon powder was used to calibrate the equipment. The type of NDAS crystallinity was determined based on the major diffraction peaks as described by Cheetham and Tao [27].

*Scanning electron microscopy (SEM).* Scanning electron micrographs of native and PDAS samples sputtered with gold to a thickness of about 30 nm (Sputter Coater Type E 5100, Biorad GmbH, Munich Germany) were taken with a DSM 940 apparatus (Carl Zeiss, Oberkochen, Germany) at a magnification of 2000X. The range of accelerating voltage was 0.3 to 30 kV. The granule's size was estimated from calibrated scale bar on the SEM micrograph [28].

*Thermal analysis.* Thermal analysis (thermogravimetric analysis (TGA), differential thermogravimetry (DTG), and Differential scanning calorimetry (DSC)) was performed using DTG (Differential Thermo Gravimetry)- 60H (SHIMADZU Corporation, Japan). Starch samples weighing in the range of 10.61 mg to 20.89 mg were used. The sample was placed in the

equipment and it was heated in a nitrogen atmosphere from 25 to 701˚C at a heating rate of 10˚C/min. Nitrogen was used as purge gas at a flow rate of 60 mL/min [29].

*Swelling power and relative solubility in water.* The swelling power and relative solubility of the native and PDAS samples in water were determined according to the method described by Garg and Jana [30]. The native and PDAS samples were dried at 60˚C for 24 h (moisture content 2.21%). The dried starch samples (0.1 g) were taken in a centrifuge tube with 10 mL of distilled water. The starch suspensions were incubated in a water bath for 1 h at different temperatures from 25 to 90˚C with a working churn. After cooling the samples to room temperature, the tubes were centrifuged at 3000 rpm for 20 min in a centrifuge (Beckman Coulter, Inc. USA) to separate the insoluble starch particles. The insoluble starch was separated from the supernatant and weighed (Wp). Both phases were dried at 105˚C for 24 h. The dry insoluble starch (Wps) and supernatant (Ws) were weighed. Swelling power was calculated as the ratio of the weight of hydrated insoluble starch granule (g)/weight of dry granule in insoluble starch (g) as shown in Eq (3).

$$Swelling\ power = \frac{Wp}{Wps} \tag{3}$$

The solubility of starch was calculated as the percentage of the dry mass of soluble in the supernatant (Ws) to the dry mass of the whole starch sample (Wo) as shown in Eq (4).

$$Solubility = \frac{Ws}{Wo} \times 100\% \tag{4}$$

*Solubility in an organic solvent.* The solubility of the NDAS and PDAS samples was determined by dispersing 50 mg of NDAS or PDAS in 10 mL of acetone, chloroform, carbon tetrachloride, dichloromethane, ethyl acetate, ethanol, or pyridine at 25˚C after 48 h [31].

**In vivo safety evaluation of PDAS in rats.** *Preparation and grouping of experimental animals.* Healthy and nonpregnant female Wistar rats with an age of 8 to 10 weeks and above and a weight of 149.12g–190.15g were obtained from the Department of Pharmacology and Clinical Pharmacy, Addis Ababa University. The animals were acclimated to laboratory conditions for 5 days. They were housed in standard cages and kept under the standard condition at a temperature of 22˚C (± 3˚C), with a 12 h light and 12 h dark cycle [32]. They were provided with free access to a standard diet and tap water ad libitum [33].

The animals were randomly assigned to a control and three treatment groups. Each animal was assigned a unique identification number. A total of 24 female rats containing 6 rats per group were used for the acute toxicity study, similarly, 24 female rats containing 6 rats per group were used for the subacute toxicity study: a control and three treatment groups [32].

*Acute toxicity study.* According to OECD guidelines [32], normal females, nulliparous, and nonpregnant rats were randomly selected and grouped into four groups (n = 6) and then kept in their cage for 5 days before dosing to allow acclimatization to the laboratory conditions. All groups of the rats fasted overnight, weighed, and doses were calculated based on their body weight. The PDAS (DS = 2.842) in 2 mL of distilled water was administered orally at a single dose of 175 mg/kg (group I), 560 mg/kg (group II), and 1792 mg/kg (group III) body weight of rats in the test groups. The control group (group IV) received 2 mL of distilled water. The animals were then kept under close observation continuously for 1 h and intermittently for 4 h, and thereafter once every 24 h for the next 14 days. During this study period, clinical observations were made for mortality, behavioral, neurological, and any other abnormalities, and their weight was measured weekly. Finally, on the 15th day, their final weights were measured, and gross physical examinations were carried out.

*Subacute toxicity study*. Twenty-four healthy adult rats were randomly distributed into four groups (I, II, III, and IV) each consisting of six female rats. Groups I, II, and III were orally administered with PDAS (DS = 2.842) in distilled water at doses of 175, 560, and 1792 mg/kg body weight per day, respectively, for 28 days using oral gavage. Group IV served as a control group and received distilled water. The clinical observation was carried out for 28 days and their weight was measured weekly for four weeks. On the 28th day, the final weight of the rats was measured [32].

## Ethical consideration

The study was conducted after having approval from the Addis Ababa University, College of Health Sciences, institutional review board (Protocol Number: 053/20/SoP*)*, and in line with the highest standard for humane and compassionate use of animals in biomedical research. Animals used in this study were not subjected to any unnecessary painful and terrifying situations [32]. The animals were protected from pathogens and placed in an appropriate environment [33].

## Statistical analysis

Statistical analysis was performed using Analysis of Variance (ANOVA) with statistical software Origin 8.5.1 (Origin LabTM Corporation, USA). Tukey multiple comparison tests were used to compare the individual difference in the physicochemical properties of the native and propylated starches. At a 95% confidence interval, p values less than or equal to 0.05 were considered statistically significant. ANOVA was used to compare the data obtained after characterization and toxicity studies. The results are reported as mean and standard deviation (SD).

## Results

### Starch yield from *Dioscorea abyssinica* tuber

The major constituent of *Dioscorea abyssinica* tuber is starch accounting for about 82.26 ± 1.57% on a dry weight basis. The native starch isolated from *Dioscorea abyssinica* tuber was pure white and tasteless with no smell.

### Determination of propionyl content and DS

Table 2 shows the variation of propionyl content and DS at different reaction times, temperatures, starch to pyridine ratios, and starch to propionic anhydride ratios. As shown in (**S1 Table**) DAS exhibited DS ranging from 0.453 to 2.842 based on the experimental conditions.

### Effect of reaction time on propionyl content and DS

Based on the data obtained as shown in Table 2, propionyl content and DS increased with an increase in reaction time. When the reaction time increased from 3 h to 24 h, propionyl content and DS of the starch increased significantly ($p < 0.05$) from 16.342% to 50.445% and 0.553 to 2.842, respectively.

### Effect of temperature on propionyl content and DS

An increase in temperature from 50˚C to 75˚C, as indicated in Table 2, resulted in significant ($p < 0.05$) increment in propionyl content and DS of the starch from 19.665% to 50.445% and 0.693 to 2.842, respectively.

**Table 2. Effect of reaction time, temperature, starch to pyridine ratio, and starch to propionic anhydride on propionyl content and DS of PDAS (n = 3, mean ± SD).**

| Reaction condition | Reaction Time (h) | Reaction Temperature (°C) | Starch: pyridine (g/mL) | Starch: propionic anhydride (g/mL) | Propionyl content (%) | DS |
|---|---|---|---|---|---|---|
| A | 3 | 75 | 1:5 | 1:8 | 16.342 ± 0.596 | 0.553 |
| B | 6 | 75 | 1:5 | 1:8 | 26.360 ±0.140 | 1.011 |
| C | 12 | 75 | 1:5 | 1:8 | 33.153± 0.715 | 1.397 |
| D | 24 | 75 | 1:5 | 1:8 | 50.445 ± 0.285 | 2.842 |
| E | 24 | 50 | 1:5 | 1:8 | 19.665± 0.285 | 0.693 |
| F | 24 | 60 | 1:5 | 1:8 | 31.445± 0.435 | 1.293 |
| G | 24 | 75 | 1:1 | 1:8 | 35.055± 0.285 | 1.520 |
| H | 24 | 75 | 1:2.5 | 1:8 | 36.497 ± 0.545 | 1.617 |
| I | 24 | 75 | 1:10 | 1:8 | 28.593± 0.081 | 1.130 |
| J | 24 | 75 | 1:5 | 1:1 | 13.775 ± 0.435 | 0.453 |
| K | 24 | 75 | 1:5 | 1:2 | 14.345 ± 0.435 | 0.474 |
| L | 24 | 75 | 1:5 | 1:4 | 37.810 ± 0.593 | 1.710 |

DS: Degree of substitution.

### Effect of starch to pyridine ratio on propionyl content and DS

As shown in Table 2, propionyl content and DS increased with an initial decrease in starch to pyridine ratios followed by a decrease in propionyl content and DS with a further decrement in starch to pyridine ratio. A decrement in starch to pyridine ratio from 1:1 to 1:5 resulted in a significant ($p < 0.05$) increment in propionyl content and DS of the starch from 35.055% to 50.445% and 1.520 to 2.842, respectively. However, decreasing the starch to pyridine ratio to 1:10 resulted in a decrement in propionyl content and DS of the starch to 28.593 and 1.130, respectively.

### Effect of starch to propionic anhydride ratio on propionyl content and DS

It was observed that DS and propionyl content of starch increased with a decrease in ratios of starch to propionic anhydride. With a decrease in the ratio of starch to propionic anhydride from 1:1 to 1:8, DS and propionyl content of starch increased significantly ($p < 0.05$) from 0.45 to 2.842 and 13.775 to 50.445%, respectively.

### Fourier Transform Infrared (FTIR) spectra

The FTIR spectra of native starch and PDAS samples with different DS are depicted in Fig 2. In the spectrum of native starch, there are bands at 1183, 1131, and 1037 cm$^{-1}$ which are attributed to C-O band stretching. There are also absorption bands at 918, 855, 746, and 577 cm$^{-1}$ due to the anhydro/glucose ring stretching vibrations. An extremely broadband due to hydrogen-bonded -OH groups appears at 3400 cm$^{-1}$. The FTIR spectra of the propylated starches show a new band at 1757 cm$^{-1}$ assigned to carbonyl C = O vibration as indicated in (S1 Fig).

### X-ray diffractograms

The X-ray diffraction patterns of NDAS and PDAS samples are presented in Fig 3. Variation in intensity of light with angle (2θ) was recorded in the diffractograms. The diffractogram of NDAS (DS = 0.0) exhibited maximum peaks at around 17° (2θ). Other significant peaks were around 14.5° (2θ), 20° (2θ), and 22° (2θ). It was observed from the diffractograms of PDAS

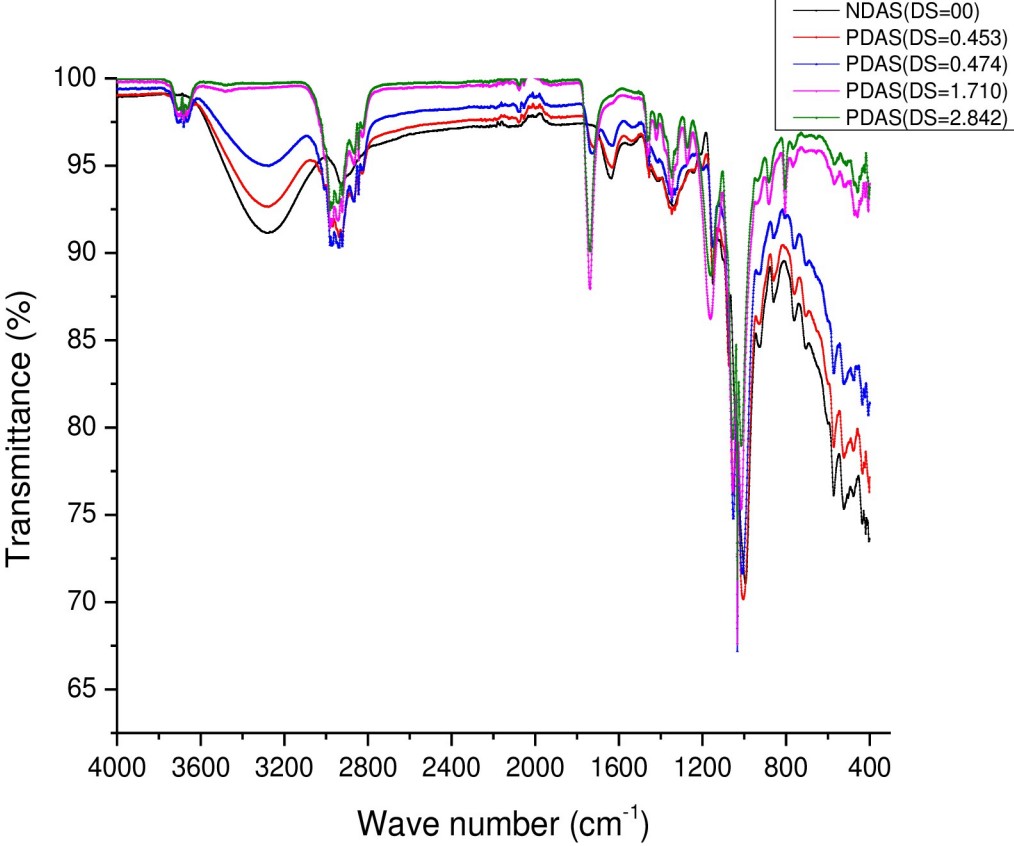

**Fig 2. Fourier Transform Infrared (FTIR) spectra of NDAS and PDAS with different DS.**

that the peaks at 14.5˚ (2θ), 20˚ (2θ), and 22˚ (2θ) disappeared after modification. New peaks developed at 9˚ and 20˚ (2θ) in PDAS with higher DS (1.710 and 2.842) as shown in (**S2 Fig**).

## Morphological properties of the starches

Scanning electron micrographs of NDAS and PDAS samples with different DS are presented in **Fig 4**. The NDAS and PDAS at lower DS (0.453 and 0.474) are composed of entirely small entities of granular structures with a rounded shape and they became irregularly shaped and form larger particles at higher DS (1.710 and 2.842).

## Thermal properties of the starches

The comparative TGA curves of the NDAS and PDAS samples at different DS are presented in **Fig 5(A)**. The thermograms of NDAS and PDAS except for low DS (DS = 0.474) showed two-stage weight loss below 421˚C. Whereas, PDAS with low DS (DS = 0.474) showed three-stage weight loss below 421˚C. The first weight loss in the temperature range of 25.82˚C to 157.74˚C was 5.56%, 4.54%, 3.92%, 1.26%, and 0.58% for DS = 00, DS = 0.453 DS = 0.474, DS = 1.710, and DS = 2.842, respectively. Once dehydrated, thermal decomposition and more pronounced weight loss in the temperature range of 258.95˚C to 420.77˚C was 47.42%, 50.62%, 42.29%, 56.18% and 49.77%for DS = 00, DS = 0.453, DS = 0.474, DS = 1.710 and DS = 2.842, respectively. However, the second stage of degradation with 70.91% weight loss at 361.97˚C was

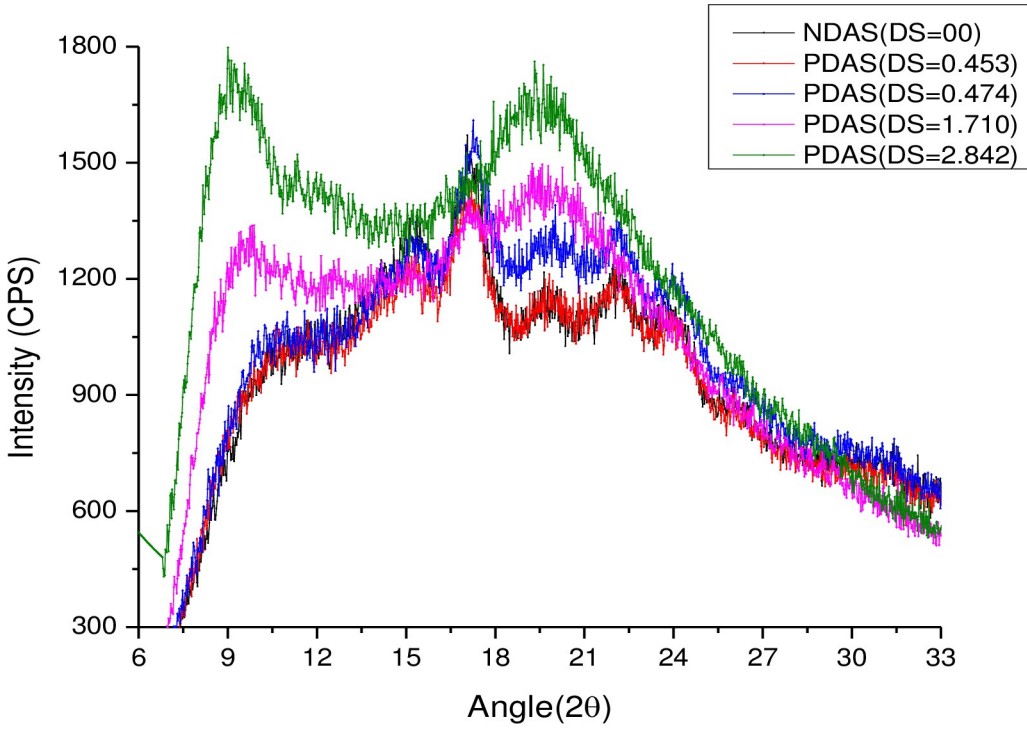

**Fig 3. X-ray diffraction patterns of NDAS and PDAS (DS = 0.453, 0.474, 1.710 and 2.842).**

observed for PDAS with lower DS (DS = 0.474). Further heating up to 701.18°C resulted in carbonization and ash formation as depicted in (**S3(A) Fig**).

The DTG thermograms of the NDAS and PDAS show one degradation peak as depicted in **Fig 5(B)**. However, PDAS with low DS (DS = 0.474) showed two-stage degradation peaks below 421°C. The Maximum degradation temperature ($T_{max}$) of the NDAS and PDAS ranged from 312.08 for DS = 00 to 373.91°C for DS = 2.842. The weight loss rate of the NDAS and PDAS ranged from 2.38%/°C for DS = 00 at 312.08°C to 2.56%/°C for (DS = 2.842) at 373.91°C as shown (**S3(B) Fig**). As shown in Table 3, the residual mass of NDAS and PDAS at 701.18°C ranged from 10.55 to 14.96 (%).

DSC thermograms of the NDAS and PDAS show peaks with an increase in temperature, the first one for moisture loss and the others for decompositions as depicted in **Fig 5(C)**. As seen in Table 4, NDAS (DS = 0.0) showed endothermic peaks with the first one corresponding to loss of water at 70.56°C with energy absorbed 461.6 J/g followed by stages of decomposition peaks at 201.43°C, 279.49°C, and 312.08°C with a total of energy absorbed 262.8 J/g. But PDAS (DS = 2.842) showed endothermic peaks with the first one corresponding to loss of water at 43.44°C with energy absorbed 59.67 J/g followed by stages of decomposition peaks at 209.05°C and 365.59°C with energy absorbed 76.43 J/g as shown in (**S3(C) Fig**).

## Swelling power and solubility of the starches

The swelling power of native and PDAS samples at low DS increased with temperature as depicted in **Fig 6**. With the increase in temperature from 25°C to 90°C, the swelling power of native starch increased significantly (p < 0.05) from 5.44 to 26 g/g. PDAS samples at low DS (DS = 0.453 and DS = 0.474) showed higher swelling power from 25°C to 80°C than native starch. But it was comparable to the native starch at 90°C. The swelling power of the

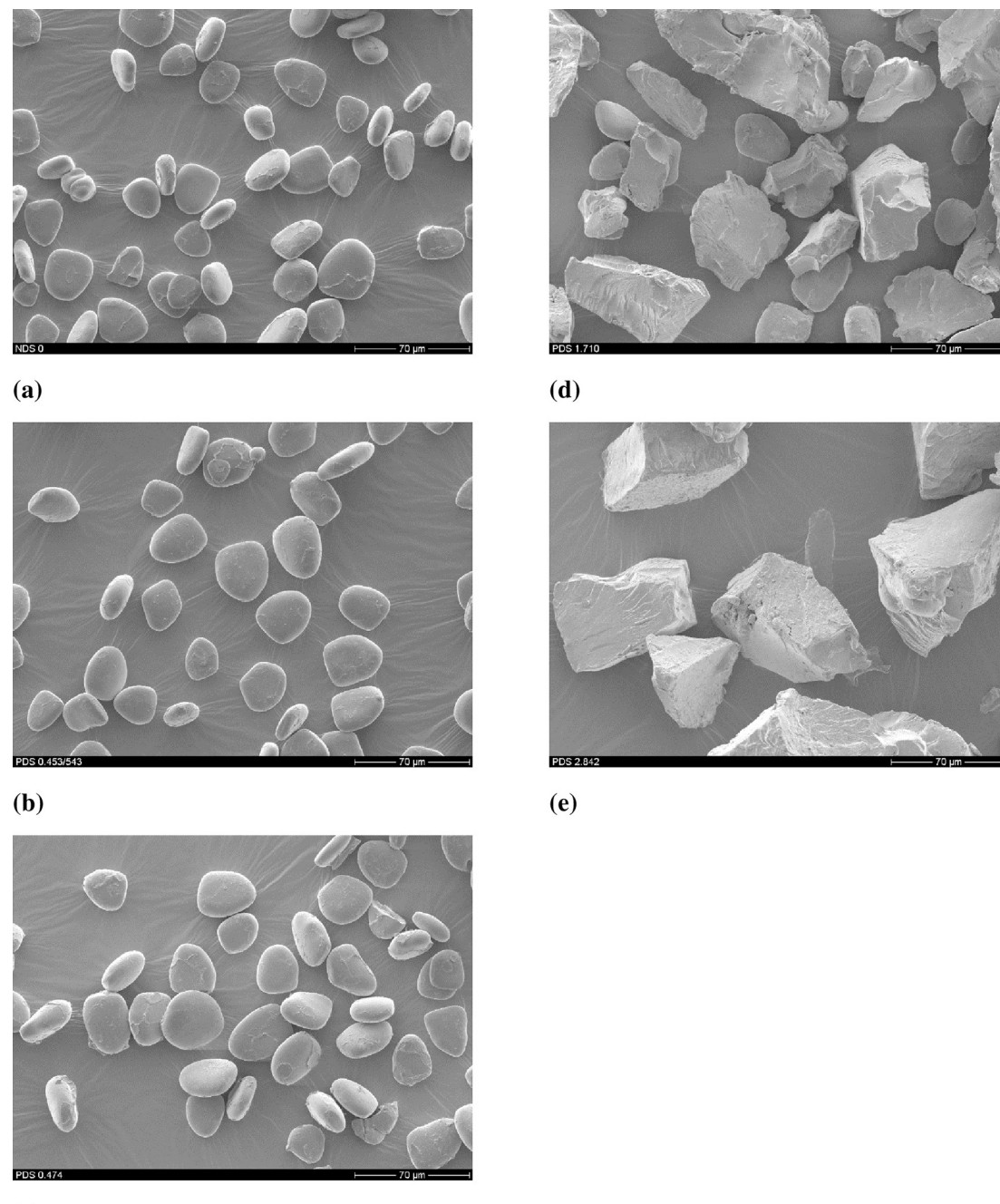

**(a)**

**(b)**

**(c)**

**(d)**

**(e)**

**Fig 4.** Scanning electron micrographs (2000X) of native DAS (a) NDAS and PDAS with different DS: (b) PDAS (DS = 0.453), (c) PDAS (DS = 0.474), (d) PDAS (DS = 1.710), (e) PDAS (DS = 2.842).

propylated starches at high DS (DS = 1.710 and DS = 2.842) was much smaller than the native starch and did not increase significantly ($p > 0.05$) with an increase in temperature as in native starch or PDAS of lower DS as depicted in (**S4 Fig**).

The relative solubility of the native and PDASs is shown in **Fig 7**. In general, the solubility of all the samples increased with increasing temperature. Solubility of propylated starches of low DS (DS = 0.453 and DS = 0.474) was higher than those of the native starch at all

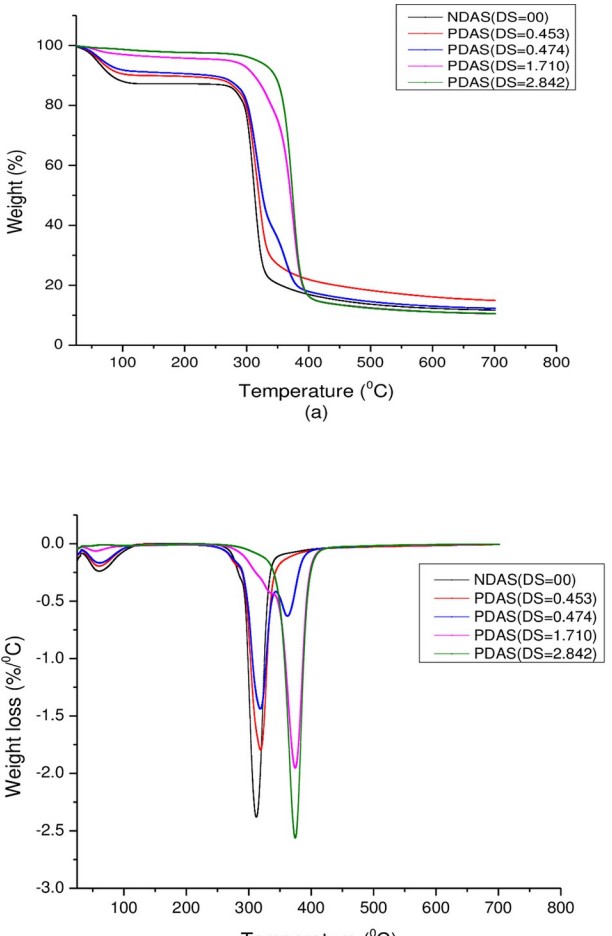

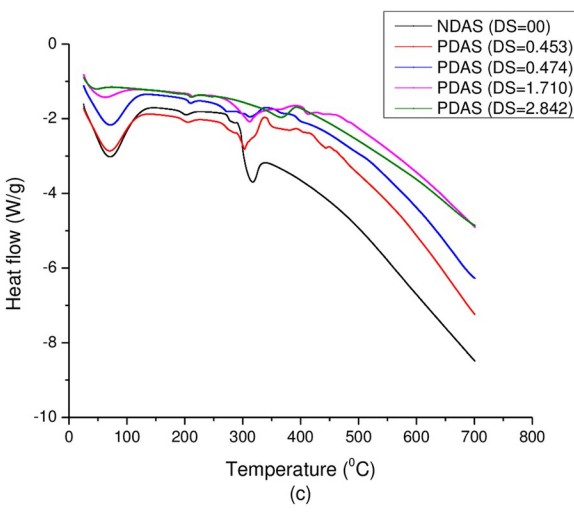

**Fig 5.** (a)TGA (b) DTG and (c) DSC thermograms of NDAS and PDAS with different DS (DS = 0.453, 0.474, 1.710 and 2.842).

**Table 3. Thermogravimetric characteristics of TGA and DTG of NDAS and PDASs with different DS (DS = 0.453, 0.474, 1.710, and 2.842).**

| No. | Type of starch | Tmax (°C) | Weight loss (%) | Weight loss rate (%/°C) | Residue at 701.18°C (%) |
|---|---|---|---|---|---|
| 1 | NDAS | 60.46 | 5.56 | -0.239 | 11.69 |
| 2 | PDAS (DS = 0.453) | 67.28 | 4.54 | -0.195 | 14.96 |
| 3 | PDAS (DS = 0.474) | 61.59 | 3.92 | -0.168 | 12.30 |
| 4 | PDAS (DS = 1.710) | 57.80 | 1.26 | -0.062 | 10.55 |
| 5 | PDAS (DS = 2.842) | 43.44 | 0.58 | -0.02 | 10.55 |

Tmax- Maximum degradation (maximum weight loss) temperature; TGA- Thermogravimetric analysis; DTG- Differential thermogravimetry; NDAS- native *Dioscorea abyssinica* starch; PDAS- propylated *Dioscorea abyssinica* starches; DS- Degree of substitution.

**Table 4. Experimental values of onset temperature (To), peak temperature (Tp), end set temperature (Te), enthalpy change (ΔH), and initial mass (mᵢ) for NDAS and PDASs with different DS (DS = 0.453, 0.474, 1.710 and 2.842).**

| No. | Material | ΔT (˚C) | To (˚C) | Tp (˚C) | Te (˚C) | ΔH (J/g) | $m_i$ (mg) |
|-----|----------|---------|---------|---------|---------|----------|-----------|
| 1 | NDAS (DS = 00) | 11.71 | 65.56 | 70.56 | 77.27 | 461.6 | 10.890 |
| 2 | PDAS (DS = 0.453) | 10.33 | 65.76 | 67.28 | 76.09 | 345.6 | 10.610 |
| 3 | PDAS (DS = 0.474) | 13.41 | 59.92 | 71.42 | 73.33 | 307.5 | 16.842 |
| 4 | PDAS (DS = 1.710) | 18.13 | 54.21 | 57.80 | 72.34 | 151.2 | 20.892 |
| 5 | PDAS (DS = 2.842) | 32.03 | 40.31 | 43.44 | 72.34 | 59.67 | 19.340 |

ΔT (Te–To)- Temperature change, To- onset temperature, Tp- peak temperature, Te- end set temperature, ΔH- enthalpy change, $m_i$-initial mass, NDAS- native *Dioscorea abyssinica* starch, PDAS- propylated *Dioscorea abyssinica* starches, DS- Degree of substitution.

temperatures while those of the propylated starches of high DS (DS = 1.710 and DS = 2.842) were much lower than the native starch as depicted in (**S5 Fig**). Moreover, the solubility decreased with an increase in DS from 1.710 to 2.842. The solubility of native starch was 12% at 90˚C while propylated starch at high DS (DS = 2.842) had solubility of only 1% at the same temperature.

## Solubility of the starches in organic solvents

Based on the data obtained as depicted in Table 5 NDAS and PDAS at lower DS (0.453 and 0.474) didn't dissolve in all types of organic solvents used. When the DS value was high (1.710 and 2.842), the PDAS was soluble in Acetone, Chloroform, Carbon tetrachloride,

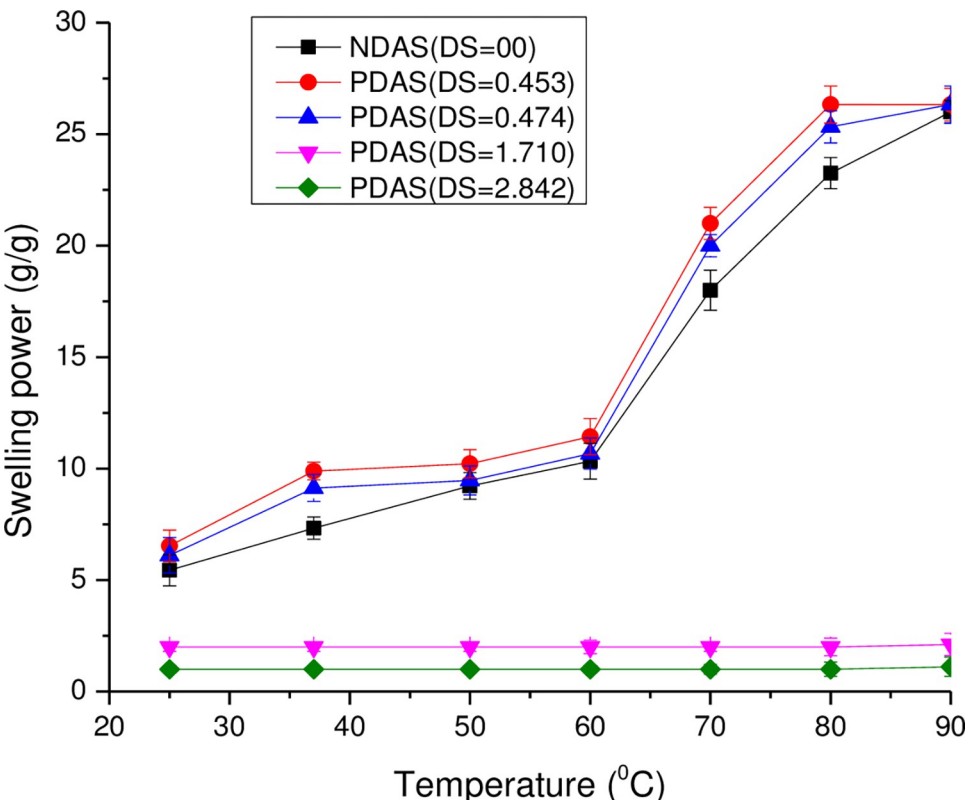

**Fig 6. Swelling power of NDAS and PDAS with different DS (DS = 0.453, 0.474, 1.710, and 2.842) as a function of temperature.**

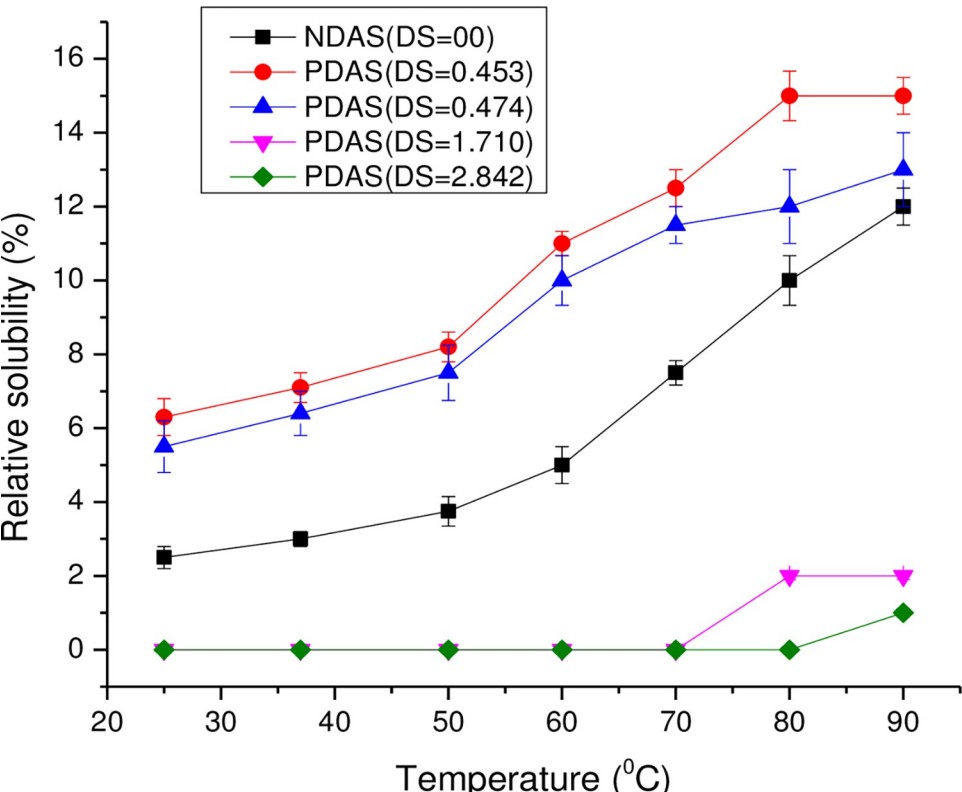

**Fig 7. Relative solubility of NDAS and PDASs with different DS (DS = 0.453, 0.474, 1.710, and 2.842) as a function of temperature.**

Dichloromethane, Ethyl acetate, and Pyridine at room temperature, and no precipitation was observed upon prolonged standing for 48 h. But the solubility of propylated starch in ethanol was not improved even at higher DS (1.710 and 2.842).

### *In vivo* safety evaluation of PDAS in Rats

**Acute toxicity of PDAS in Rats.** There was a gradual increase in the body weight of both the treated and control rats as depicted in Table 6. The initial mean body of control rats was

**Table 5. Solubility of NDAS and PDASs with different DS (DS = 0.453, 0.474, 1.710, and 2.842) in organic solvents at 25˚C.**

| Type of organic solvent | Solubility | | | | |
|---|---|---|---|---|---|
| | NDAS | PDAS | | | |
| | | DS = 0.453 | DS = 0.474 | DS = 1.710 | DS = 2.842 |
| Carbon tetrachloride | – | – | – | ++ | ++ |
| Chloroform | – | – | – | ++ | ++ |
| Ethyl acetate | – | – | – | ++ | ++ |
| Dichloromethane | – | – | – | ++ | ++ |
| Pyridine | – | – | – | ++ | ++ |
| Acetone | – | – | – | ++ | ++ |
| Ethanol | – | – | – | – | – |

DS: Degree of substitution, NDAS: Native *Dioscorea abyssinica* starch, PDAS: Propylated *Dioscorea abyssinica* starch, –: Insoluble, ++: Soluble.

**Table 6. Effect of PDAS (DS = 2.842) in distilled water on body weight increment of treated and control rats during acute toxicity study.**

| Group | Body weight in grams, n = 6 (mean ± SD) | | |
|---|---|---|---|
| | Initial | 1st week | 2nd week |
| I | 174.55 ± 7.66 | 179.90 ± 7.14 | 183.57 ± 6.85 |
| II | 166.43 ± 12.06 | 171.06 ± 12.61 | 175.67 ± 11.82 |
| III | 159.10 ± 4.34 | 163.06 ± 4.65 | 167.66 ± 4.36 |
| IV | 165.68 ± 4.39 | 170.04 ± 4.15 | 173.79 ± 3.96 |

The dose received by treatment group I, II, and III were 175mg/kg, 560 mg/kg, and 1792 mg/kg respectively, whereas group IV was used as a control group.

165.68 ± 4.39g; at the end of the experiment, their final mean body weight was 173.79 ± 3.96g. The mean body weight gain for the control rats was 8.11 g. The initial mean body weights of rats treated with doses of 175 mg/kg, 560 mg/kg, and 1792 mg/kg body weight of PDAS (DS = 2.842) were 174.55 ± 7.66 gm,166.43 ± 12.06 g, and 159.10 ± 4.34g, respectively.

In acute toxicity studies, healthy, nulliparous, and nonpregnant female Wistar rats, that received up to the dose of 1792 mg/Kg body weight of PDAS (DS = 2.842) did not show any lethal effects and toxicity symptoms like increased motor activity, ptosis, tonic extension, lacrimation, Straub reactions, exophthalmos, piloerection, salivation, muscle spasm, opisthotonus, writhing, loss on righting reflex, depression, ataxia, stimulation, sedation, hypnosis, cyanosis, analgesia. At the end of the two-week acute toxicity study, the final mean body weight of rats treated with 175 mg/kg, 560 mg/kg, and 1792 mg/kg body weight of PDAS (DS = 2.842) was 183.57 ± 6.85gm, 175.67 ± 11.82 g, and 167.66 ± 4.36 g, respectively. The mean body weight gains for rats treated with 175 mg/kg, 560 mg/kg, and 1792 mg/kg were 9.02g, 9.24 g, and 8.56 g, respectively as shown in (S2 Table).

**Sub-acute toxicity of PDAS in rats.** During the subacute experimental period, all groups of rats showed a gradual and normal increase in their body weight as shown in **Fig 8**. The

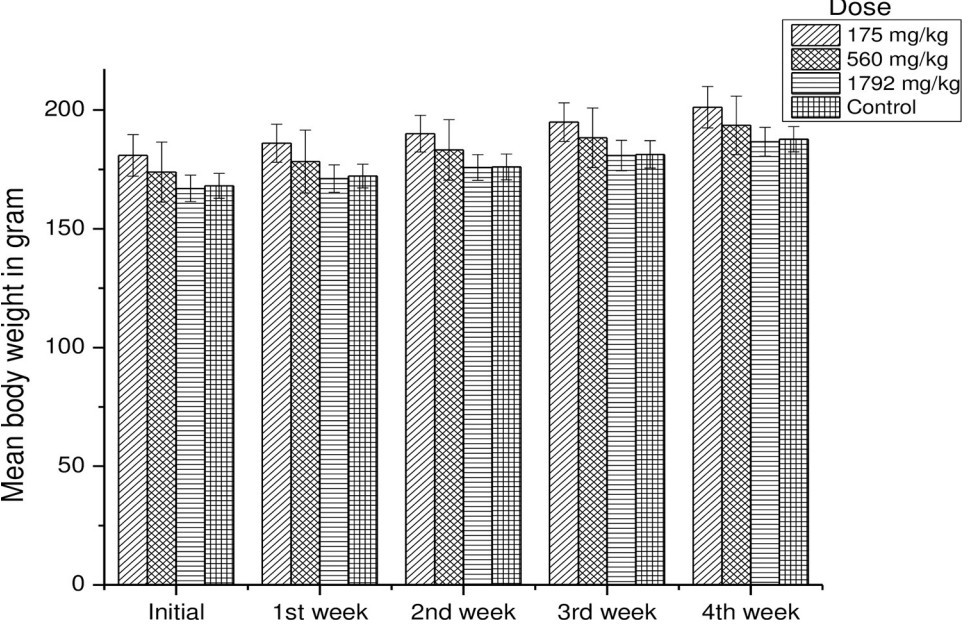

**Fig 8. Mean body weight change in rats treated with 175 mg/kg, 560 mg/kg, and 1792 mg/kg of PDAS (DS = 2.842) in distilled water as compared to the control group during the subacute toxicity study.**

initial mean bodyweight of the control group was 168.09 ± 5.24 gm, and the final mean body weight was 187.75 ± 5.32 gm. The initial mean body weight of rats treated with the dose of 175 mg/kg, 560 mg/kg, and 1792 mg/ kg of body weight of PDAS (DS = 2.842) was 180.97 ± 8.77 g, 173.90 ± 12.68 g, and 167.01 ± 5.57 g, respectively. The final mean body weight of rats treated with the dose of 175 mg/kg, 560 mg/kg, and 1792 mg/ kg of body weight of PDAS (DS = 2.842) was 201.22 ± 8.74 g, 193.60 ± 12.32 g, and 186.71 ± 6.05 g, respectively as shown in (**S6 Fig**).

Throughout the study period, no sign of toxicity or mortality was observed in treated rats, which received 175 mg/kg, 560 mg/kg, and 1792 mg/kg of body weight of PDAS (DS = 2.842). During the subacute toxicity study, the treated rats showed a gradual normal increase in their body weight. There was no significant difference in mean body weight gain of control, group I (175 mg/kg), group II (560 mg/kg), and group III (1792 mg/kg), which were 19.66 g, 20.97 g, 19.7 g, and 19.7 g, respectively.

## Discussion

The physicochemical properties of *Dioscorea abyssinica* and its proximate compositions are reported in a previous work by Gebre-Mariam and Schmidt [2]. Propylated starch samples were prepared with a different DS. Addition–elimination mechanism is responsible for the acylation of starch. The reactivities of the three free OH groups of the starch are different. As a result of steric hindrance, the secondary OH groups on $C_{(2)}$ and $C_{(3)}$ are less reactive than the primary OH group on $C_{(6)}$. The $C_{(3)}$ OH is less reactive than the $C_{(2)}$ OH. because the latter is closer to the hemiacetal group and more acidic than the former [34–36].

The increase in the DS with time is the result of an enhanced period of contact between the esterifying reagent (propionic anhydride) and the starch molecules. A longer reaction time also enhances starch swelling and ultimately improves the homogeneity of the reactants. These observations are consistent with the report of Lawal et al. [36]. An increase in temperature enhances the solubility of the esterifying agents and also facilitates the diffusion of the reactants. These observations are similar to those of Nwokocha and Ogumola [37]. Pyridine was used to provide the necessary alkaline environment for the propylation process and as a swelling agent to facilitate diffusion and penetration of the esterifying agent to the starch granular structure. The decline in DS as the concentration of pyridine increased could be attributed to the competing reactions between pyridine and starch for propionic anhydride during the propylation process. Further increase in pyridine concentration above 1:5.0 ratio causes a side reaction of propionic anhydride with pyridine and hence it is consumed before it reacts with starch [36,37].

At high ratios of starch to propionic anhydride, the numbers of propionyl groups are relatively few to sufficiently convert the hydroxyl groups. However, at low ratios of starch to propionic anhydride, the propionyl groups sufficiently acylate the hydroxyl groups resulting in higher DS. As the reaction was not a homogeneous one, it is likely that some glucose units were completely inaccessible to the propionic anhydride and so had unacylated OH groups. As a result, DS was less than 3. This replacement of the hydrophilic hydroxyl groups of the native starch by the bulk hydrophobic acyl groups reduces its hydrophilicity and renders the starch hydrophobicity [38,39]. This is in agreement with the solubility of PDAS with high DS (1.710 and 2.842) in different organic solvents used in this study.

The new absorption band in the FTIR study suggests that PDAS products were formed because of acylation. With an increase in DS of PDAS samples from 0.453 to 2.842, the intensity of the peak at 3400 cm$^{-1}$ weakened and almost disappeared at DS 1.710 and 2.842, indicating that hydroxyl groups participated in the reaction. This is in agreement with the findings of Garg and Jana [25].

Starch is a mixture of linear amylose and branched amylopectin and a semicrystalline polymer. The linear amylose, composed of α-1,4-glucopyranose bonds, is responsible for the amorphous region, while the large amylopectin contributes to the crystalline region. The highly ordered crystalline structure is due to the intramolecular and intermolecular hydrogen bonds. The propionyl groups in the modified starch samples replaced most of the hydroxyl groups of native starch, destroying the ordered crystalline structure [40,41].

Starch could adopt different crystalline structures (A, B, C, and V) [42,43]. The A, B, and C polymorphs consist of packed double helix structures. The V-type is a single helix structure. The A-type is largely present in cereal starches and B type in potato, amylose maize, and retrograded starches. Each type of structure gives typical characteristic peaks described by various researchers [44,45]. Chi *et al.*, [44]. described that the B pattern of potato starch exhibited sharp peaks at 15˚ (2θ), 17˚ (2θ), 18˚ (2θ), and 23˚ (2θ). A maximum peak at around 17˚ (2θ) following an X-ray diffraction study on NDAS is observed indicating that the crystalline structure is B-type as reported by Gebre-Mariam and Schmidt [2]. X-ray diffraction of NDAS and PDAS samples demonstrated that the crystalline structure of NDAS was lost after propylation. The disappearance of the peaks was directly proportional to the DS which is in agreement with the results reported by Garg and Jana [25].

Granule size and shape are related to the biological source from which the starch is isolated. As depicted in Fig 4, the morphology of NDAS and PDAS at lower DS (0.453 and 0.474) is changed from small entities of granular structures with a rounded shape to irregularly shaped and larger particles at higher DS (1.710 and 2.842), as reported by Garg and Jana [30].

TGA, DTG, and DSC curves reveal that all the PDAS samples decompose at higher temperatures compared to the native starch. The PDAS was thermally more stable than the native starch. Covalent bonding of the–OH groups, molecular weight increment, and decrement in the amount of -OH groups following propylation of the starch molecules is responsible for the increase in thermal stability of PDAS with an increase in DS. The main product of decomposition below 300˚C was the formation of water by intermolecular or intramolecular condensation of starch hydroxyls. The NDAS and PDAS are thermally decomposed, carbonized, and form ash when the temperature is above 300˚C [46]. In a study reported elsewhere, the thermal properties of starch succinates showed that the fewer the number of hydroxyl groups remained, the better the thermal stability of the starch esters [47].

The starch granules interact with water at high temperatures. But at low temperatures, starch solubility is limited due to large and tightly bound micellar structures but increases dramatically at higher temperatures. Starch molecules were probably more thermodynamically active as the medium temperature rise. As a result, granular mobility improves possibly, due to macromolecular instability and higher water penetration [48]. Furthermore, when starch is cooked in excess water, the crystalline structure is disturbed, and water molecules are attached to the exposed hydroxyl groups of amylose and amylopectin via hydrogen bonding resulting in an increased granule solubility [49].

At a low level of DS, the acyl groups were not sufficient to change the behavior of the hydroxyl groups. There was the weakening of intermolecular hydrogen bonds in starch with the introduction of propionyl groups. At high DS, the propionyl groups replaced most of the hydroxyl groups on starch, and interaction with water decreased. Therefore, at high DS, swelling power and solubility decreased to below that of native starch. Results showed that swelling power and solubility depended on two opposing effects: (a) the opening up of the starch structure at a low level of acylation, rendering it more accessible to water, and (b) the increasing hydrophobic character of polymer chains which gradually become the predominant effect with an increase in the DS. The introduction of acetyl groups into polymer chains resulted in the destabilization of the granular structure of starch. At high DS, the difference in the degree

of availability of water binding sites among the starch samples contributed to the variation in swelling power and solubility [16,50].

The solubility of acylated starch is dependent on the extent of acylation, degree of degradation of the acylated molecule, degree of polymerization of the ester, the fractionation of the starch or the derivative, the nature of the acyl substituent, the type of starch, temperature and solvent type [51]. When the DS value was high (1.710 and 2.842), the PDAS was soluble in Acetone, Chloroform, Carbon tetrachloride, Dichloromethane, Ethyl acetate, and Pyridine as shown in Table 5, due to the high DS on the starch surface containing hydrophobic groups. The hydrogen bonds are disrupted by the hydrophobic methyl groups and the solubility was enhanced by the propionyl groups. The solubility analysis indicates that propylation dramatically changed the solubility of the starch in organic solvents [31].

Body weight change is an important index for the assessment of toxicity [52]. In the present study, the weight gain difference between the control and treatment groups was statistically insignificant ($p > 0.05$). From the acute toxicity results, it can be concluded that the prepared PDAS (DS = 2.842) is non-toxic and safe. The maximum dose of 1792 mg/Kg body weight was below $LD_{50}$ of orally administered PDAS (DS = 2.842). Similar results were reported by Kumar and Mudili, for starch glutamate [53].

Increment in body weight determines the positive health status of the animals [54]. Subacute toxicity study examines toxicity caused by repeated dosing over an extended period of 28 days of oral administration in rodents. This test provides information on the potential of the test chemical to accumulate in the organism and which was then used as the basis for the determination of the no observed effect level (NOEL) [55].

## Conclusions

Starch was modified with propionic anhydride at different reaction conditions to synthesize and characterize PDAS. The FTIR spectra of the PDAS confirm the replacement of hydroxyl groups of the NDAS starch by bulk propionyl groups in PDAS. X-ray diffraction analysis demonstrated that the B-type crystalline structure of the NDAS starch got lost due to propylation. Scanning electron micrographs revealed that the small rounded granular structure of NDAS is gradually changed into irregularly shaped structures following propylation. Thermogravimetric studies revealed that all the PDAS samples were more thermally stable than the native starch. The swelling power, solubility in water, and organic solvents were controlled by the DS. Changes in physicochemical properties of PDAS at high DS confirmed the hydrophobic transformation in NDAS. The acute toxicity study of the PDAS (DS = 2.842) in distilled water did not produce adverse effects at doses administered orally. The maximum dose of 1792 mg/Kg body weight was below $LD_{50}$ of orally administered PDAS (DS = 2.842). Meanwhile, a subacute toxicity study of the PDAS (DS = 2.842) in distilled water did not adversely affect the body weight and behavioral parameters in the rats. Hence, it can be concluded that PDAS can be considered a generally safe excipient and fulfills physicochemical properties of a hydrophobic carrier.

## Supporting information

**S1 Fig. Fourier Transform Infrared (FTIR) spectra of NDAS and PDAS with different DS.** (RAR)

**S2 Fig. X-ray diffraction patterns of NDAS and PDAS (DS = 0.453, 0.474, 1.710 and 2.842).** (RAR)

**S3 Fig.** (a)TGA (b) DTG and (c) DSC thermograms of NDAS and PDAS with different DS (DS = 0.453, 0.474, 1.710 and 2.842).
(RAR)

**S4 Fig. Swelling power of NDAS and PDAS with different DS (DS = 0.453, 0.474, 1.710, and 2.842) as a function of temperature.**
(DOCX)

**S5 Fig. Relative solubility of NDAS and PDAS with different DS (DS = 0.453, 0.474, 1.710, and 2.842) as a function of temperature.**
(DOCX)

**S6 Fig. Mean body weight change in rats treated with 175 mg/kg, 560 mg/kg, and 1792 mg/kg of PDS (DS = 2.842) in distilled water as compared to the control group during sub-acute toxicity study.**
(DOCX)

**S1 Table. Effect of reaction time, temperature, starch to pyridine ratio and starch to propionic anhydride on propionyl content and DS of PDAS (n = 3, mean ± SD).**
(DOCX)

**S2 Table. Effect of PDAS (DS = 2.842) in distilled water on body weight increment of treated and control rats during acute toxicity study.**
(DOCX)

## Acknowledgments

The authors are grateful to the Department of Pharmaceutics and Social Pharmacy, School of Pharmacy, College of Health Sciences, Addis Ababa University; Department of Chemistry, College of Natural Sciences, Addis Ababa University; Institute of Applied Dermatopharmacy at the Martin-Luther University Halle-Wittenberg, Germany and Cadila Pharmaceuticals PLC for providing access to their laboratory facilities.

## Author Contributions

**Conceptualization:** Yonas Brhane, Tsige Gebre-Mariam, Anteneh Belete.

**Data curation:** Yonas Brhane, Tsige Gebre-Mariam, Anteneh Belete.

**Formal analysis:** Yonas Brhane, Tsige Gebre-Mariam, Anteneh Belete.

**Funding acquisition:** Yonas Brhane.

**Investigation:** Yonas Brhane.

**Methodology:** Yonas Brhane.

**Project administration:** Yonas Brhane.

**Resources:** Yonas Brhane.

**Software:** Yonas Brhane.

**Supervision:** Tsige Gebre-Mariam, Anteneh Belete.

**Validation:** Yonas Brhane, Tsige Gebre-Mariam, Anteneh Belete.

**Visualization:** Yonas Brhane, Tsige Gebre-Mariam, Anteneh Belete.

**Writing – original draft:** Yonas Brhane, Tsige Gebre-Mariam, Anteneh Belete.

**Writing – review & editing:** Yonas Brhane, Tsige Gebre-Mariam, Anteneh Belete.

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
