## [Decision Letter · Decision Letter 0]

18 Aug 2022

PONE-D-22-13047Synthesis, characterization and in vivo safety evaluation of propylated Dioscorea   abyssinica starchPLOS ONE

Dear Dr. Gabre-Mariam,

Thank you for submitting your manuscript to PLOS ONE. After careful consideration, we feel that it has merit but does not fully meet PLOS ONE’s publication criteria as it currently stands. Therefore, we invite you to submit a revised version of the manuscript that addresses the points raised during the review process.

We look forward to receiving your revised manuscript.

Kind regards,

Amjad Khan, Pharm-D; PhD

Academic Editor

PLOS ONE

Journal Requirements:

One of the authors (Y.B.) would like to thank the Center for Innovative Drug Development and Therapeutic Trials for Africa, for sponsoring the work

Additional Editor Comments:

Changes are recommended by the reviewers to be incorporated

Reviewers' comments:

Reviewer's Responses to Questions

**Comments to the Author**

1. Is the manuscript technically sound, and do the data support the conclusions?

Reviewer #1: Yes

Reviewer #2: Yes

2. Has the statistical analysis been performed appropriately and rigorously? 

Reviewer #1: Yes

Reviewer #2: Yes

3. Have the authors made all data underlying the findings in their manuscript fully available?

Reviewer #1: Yes

Reviewer #2: Yes

4. Is the manuscript presented in an intelligible fashion and written in standard English?

Reviewer #1: Yes

Reviewer #2: Yes

5. Review Comments to the Author

Reviewer #1: Comments

The manuscript titled “Synthesis, characterization and in vivo safety evaluation of propylated Dioscorea abyssinica starch” submitted to plos one was thoroughly reviewed. The study relates to evaluation of propylated Dioscorea abyssinica starch as a pharmaceutical adjuvant. Author propylated the isolated starch and then evaluated it for Crystallinity, morphological structure, thermal behavior, solubility, and safety. Development of natural product based material is need of the day due to their safety. It is a good study and conducted in a proper way.

Here are some of my observations/ suggestions;

• Introduction is not properly arranged and needs to be made in line with the theme of manuscript. P#3, First paragraph: Same references are repeated again again within the same paragraph, indicating that the text has been taken from a single source.

• Purity level of the chemicals needs to be mentioned. It should also be stated that they were used as received or further processing was done.

• It needs to state that in which mode (absorbance or transmittance) was FTIR spectra obtained. The operating software may also be mentioned

• Tables are not of uniform format

• Conclusion needs to be concise. It has unnecessary text which should be omitted

• In some references page numbers are missing which should be included

Recommendation:

I suggest to accept the manuscript after incorporation of the suggested changes.

Reviewer #2: This manuscript is clearly written and well organized. The introduction and background are quite reasonable. The manuscript is accepted with minor revision. There are some observations which are submitted as attachment that needs to be addressed by the author.

There are multiple spelling and grammatical errors throughout the entire manuscript which needs to be correctedThe objective and need of the study should be clearly mentionedThe source and purity of each chemical used during the study should be addedList of instruments used during the study should be providedUnit of milli liter is mL. In the text, it is mentioned as ml in multiple places throughout the manuscript and needs to be correctedMethods for Starch extraction from Dioscorea abyssinica tuber should be sated completelyEach of the method used during the study should be stated completely and referenced properlyThe level or degree of decimal given in data in tables and text should be uniform in the table and throughout the manuscriptThe references of manuscript should be properly arrange as per journal requirement   ********** 

6. PLOS authors have the option to publish the peer review history of their article (what does this mean?). If published, this will include your full peer review and any attached files.

Reviewer #1: No

Reviewer #2: **Yes: **Dr. Farhad Ullah

---

## [Author Response · Author response to Decision Letter 0]

25 Aug 2022

PONE-D-22-13047

Synthesis, characterization, and in vivo safety evaluation of propylated Dioscorea abyssinica starch

Editor comment #1: Please ensure that your manuscript meets PLOS ONE's style requirements, including those for file naming. The PLOS ONE style templates can be found at 

Author response to editor comment #1: The manuscript is now revised based on PLOS ONE style templates

Editor comment #2: Thank you for stating the following financial disclosure: 

One of the authors (Y.B.) would like to thank the Center for Innovative Drug Development and Therapeutic Trials for Africa, for sponsoring the work

Author response to editor comment #2: This statement is incorporated in the cover letter.

Editor comment #3: Please review your reference list to ensure that it is complete and correct. If you have cited papers that have been retracted, please include the rationale for doing so in the manuscript text, or remove these references and replace them with relevant current references. Any changes to the reference list should be mentioned in the rebuttal letter that accompanies your revised manuscript. If you need to cite a retracted article, indicate the article’s retracted status in the References list and also include a citation and full reference for the retraction notice.

Author response to editor comment #3: The reference list is reviewed and the following references (5, 6, 32, 33, 35, 43, 52, 54, 55) are modified by incorporating missing page numbers.

Editor comment #4: Changes are recommended by the reviewers to be incorporated.

Author response to editor comment #4: The manuscript is revised; all the comments given by the reviewers have been accommodated.

Comments to the Author

1. Is the manuscript technically sound, and do the data support the conclusions?

Reviewer #1: Yes

Reviewer #2: Yes

2. Has the statistical analysis been performed appropriately and rigorously?

Reviewer #1: Yes

Reviewer #2: Yes

3. Have the authors made all data underlying the findings in their manuscript fully available?

Reviewer #1: Yes

Reviewer #2: Yes

4. Is the manuscript presented in an intelligible fashion and written in standard English?

Reviewer #1: Yes

Reviewer #2: Yes

5. Review Comments to the Author

Reviewer #1: Comments

The manuscript titled “Synthesis, characterization and in vivo safety evaluation of propylated Dioscorea abyssinica starch” submitted to plos one was thoroughly reviewed. The study relates to evaluation of propylated Dioscorea abyssinica starch as a pharmaceutical adjuvant. Author propylated the isolated starch and then evaluated it for Crystallinity, morphological structure, thermal behavior, solubility, and safety. Development of natural product-based material is need of the day due to their safety. It is a good study and conducted in a proper way.

Here are some of my observations/ suggestions;

Reviewer #1: Comment 1: Introduction is not properly arranged and needs to be made in line with the theme of manuscript. P#3, First paragraph: Same references are repeated again within the same paragraph, indicating that the text has been taken from a single source.

Author response to Reviewer #1: Comment 1: The statement that is not related with the theme of manuscript is removed. 

Reviewer #1: Comment 2: Purity level of the chemicals needs to be mentioned. It should also be stated that they were used as received or further processing was done.

Author response to Reviewer #1: Comment 2: The comment is accommodated in the revised manuscript.

Reviewer #1: Comment 3: It needs to state that in which mode (absorbance or transmittance) was FTIR spectra obtained. The operating software may also be mentioned

Author response to Reviewer #1: Comment 3: FTIR spectra were obtained with transmittance mode and data acquired from FTIR spectrophotometer (PerkinElmer, Spectrum Two DTGS, UK) were plotted in Origin Pro 8.5.1 software. This is stated in the revised version of the manuscript.

Reviewer #1: Comment 4: Tables are not of uniform format.

Author response to Reviewer #1: Comment 4: The format of the tables is made uniform.

Reviewer #1: Comment 5: Conclusion needs to be concise. It has unnecessary text which should be omitted.

Author response to Reviewer #1: Comment 5: The conclusion is made precise. 

Reviewer #1: Comment 6: In some references page numbers are missing which should be included.

Author response to Reviewer #1: Comment 6: The missing page numbers of the following references (5, 6, 32, 33, 35, 43, 52, 54, 55) are incorporated. 

Recommendation by Reviewer #1:

I suggest to accept the manuscript after incorporation of the suggested changes.

Reviewer #2: Comments

This manuscript is clearly written and well organized. The introduction and background are quite reasonable. The manuscript is accepted with minor revision. There are some observations which are submitted as attachment that needs to be addressed by the author.

Reviewer #2: Comment 1: There are multiple spelling and grammatical errors throughout the entire manuscript which needs to be corrected.

Author response to Reviewer #2: Comment 1: The spelling and grammatical errors have been corrected.

Reviewer #2: Comment 2: The objective and need of the study should be clearly mentioned.

Author response to Reviewer #2: Comment 2: The objective of this study is now stated as “The objective of this research work is to develop an effective propylated Dioscorea abyssinica starch (PDAS) as a hydrophobic excipient for pharmaceutical applications with a reasonable price.”

Reviewer #2: Comment 3: The source and purity of each chemical used during the study should be added

Author response to Reviewer #2: Comment 3: Comment addressed.

Reviewer #2: Comment 4: List of instruments used during the study should be provided.

Author response to Reviewer #2: Comment 4: The instruments used are specified in the method section. But, the list of instruments is not prepared separately as it is not the requirement of PLOS ONE.

Reviewer #2: Comment 5: Unit of milli liter is mL. In the text, it is mentioned as ml in multiple places throughout the manuscript and needs to be corrected.

Author response to Reviewer #2: Comment 5: Corrected. 

Reviewer #2: Comment 6: Methods for Starch extraction from Dioscorea abyssinica tuber should be stated completely.

Author response to Reviewer #2: Comment 6: The extraction method is stated completely.

Reviewer #2: Comment 7: Each of the method used during the study should be stated completely and referenced properly.

Author response to Reviewer #2: Comment 7: All the methods are stated completely and properly referenced.

Reviewer #2: Comment 8: The level or degree of decimal given in data in tables and text should be uniform in the table and throughout the manuscript.

Author response to Reviewer #2: Comment 8: The uniformity of significant figures is now maintained.

Reviewer #2: Comment 9: The references of manuscript should be properly arranged as per journal requirement 

Author response to Reviewer #2: Comment 9: The references of the manuscript have been prepared as per the Journal’s requirement.

6. PLOS authors have the option to publish the peer review history of their article (what does this mean?). If published, this will include your full peer review and any attached files.

If you choose “no”, your identity will remain anonymous but your review may still be made public

Do you want your identity to be public for this peer review? For information about this choice, including consent withdrawal, please see our Privacy Policy.

Reviewer #1: No

Reviewer #2: Yes: Dr. Farhad Ullah

---

## [Decision Letter · Decision Letter 1]

18 Oct 2022

Synthesis, characterization, and in vivo safety evaluation of propylated Dioscorea abyssinica starch

PONE-D-22-13047R1

Dear Dr. Tsige Gebre-Mariam,

We’re pleased to inform you that your manuscript has been judged scientifically suitable for publication and will be formally accepted for publication once it meets all outstanding technical requirements.

Kind regards,

Amjad Khan, Pharm-D; PhD

Academic Editor

PLOS ONE

Additional Editor Comments (optional):

Reviewers' comments:

Reviewer's Responses to Questions

**Comments to the Author**

1. If the authors have adequately addressed your comments raised in a previous round of review and you feel that this manuscript is now acceptable for publication, you may indicate that here to bypass the “Comments to the Author” section, enter your conflict of interest statement in the “Confidential to Editor” section, and submit your "Accept" recommendation.

Reviewer #2: All comments have been addressed

2. Is the manuscript technically sound, and do the data support the conclusions?

Reviewer #2: Yes

3. Has the statistical analysis been performed appropriately and rigorously? 

Reviewer #2: Yes

4. Have the authors made all data underlying the findings in their manuscript fully available?

Reviewer #2: Yes

5. Is the manuscript presented in an intelligible fashion and written in standard English?

Reviewer #2: Yes

6. Review Comments to the Author

Reviewer #2: This manuscript is clearly written and well organized. All the suggested changes have been incorporated. The manuscript is acceptable in present form.

7. PLOS authors have the option to publish the peer review history of their article (what does this mean?). If published, this will include your full peer review and any attached files.

Reviewer #2: No

---

## [Editor Report · Acceptance letter]

16 Nov 2022

PONE-D-22-13047R1 

Synthesis, characterization, and *in vivo* safety evaluation of propylated *Dioscorea abyssinica* starch

Dear Dr. Gebre-Mariam:

I'm pleased to inform you that your manuscript has been deemed suitable for publication in PLOS ONE. Congratulations! Your manuscript is now with our production department. 

Kind regards, 

on behalf of

Dr. Amjad Khan 

Academic Editor

PLOS ONE